# Mitochondrial PCK2 Missense Variant in Shetland Sheepdogs with Paroxysmal Exercise-Induced Dyskinesia (PED)

**DOI:** 10.3390/genes11070774

**Published:** 2020-07-09

**Authors:** Jasmin Nessler, Petra Hug, Paul J. J. Mandigers, Peter A. J. Leegwater, Vidhya Jagannathan, Anibh M. Das, Marco Rosati, Kaspar Matiasek, Adrian C. Sewell, Marion Kornberg, Marina Hoffmann, Petra Wolf, Andrea Fischer, Andrea Tipold, Tosso Leeb

**Affiliations:** 1Department of Small Animal Medicine and Surgery, University of Veterinary Medicine Hannover Foundation, 30559 Hannover, Germany; jasmin.nessler@tiho-hannover.de (J.N.); andrea.tipold@tiho-hannover.de (A.T.); 2Institute of Genetics, Vetsuisse Faculty, University of Bern, 3001 Bern, Switzerland; petrahug@bluewin.ch (P.H.); vidhya.jagannathan@vetsuisse.unibe.ch (V.J.); 3Department of Clinical Sciences, Faculty of Veterinary Medicine, Utrecht University, 3584 CM Utrecht, The Netherlands; p.j.j.mandigers@veterinair-neuroloog.nl (P.J.J.M.); P.A.J.Leegwater@uu.nl (P.A.J.L.); 4Department of Pediatrics, Hannover Medical School, 30625 Hannover, Germany; Das.Anibh@mh-hannover.de; 5Section of Clinical and Comparative Neuropathology, Institute of Veterinary Pathology at the Centre for Clinical Veterinary Medicine, Ludwig-Maximilians-Universität, 80539 Munich, Germany; marco.rosati@neuropathologie.de (M.R.); kaspar.matiasek@neuropathologie.de (K.M.); 6Biocontrol, Labor für Veterinärmedizinische Diagnostik, 55218 Ingelheim, Germany; a.sewell@freenet.de; 7AniCura Tierklinik Trier GbR, 54294 Trier, Germany; Kornberg@t-online.de; 8Tierklinik Stommeln, 50259 Puhlheim, Germany; dr.marinahoffmann@googlemail.com; 9Nutritional Physiology and Animal Nutrition, University of Rostock, 18059 Rostock, Germany; petra.wolf@uni-rostock.de; 10Section of Neurology, Clinic of Small Animal Medicine, Ludwig-Maximilians-Universität, 80539 Munich, Germany; A.Fischer@medizinische-kleintierklinik.de

**Keywords:** *Canis lupus familiaris*, whole genome sequencing, dog, mitochondrion, phosphoenolpyruvate-carboxykinase, inborn error of metabolism, precision medicine

## Abstract

Four female Shetland Sheepdogs with hypertonic paroxysmal dyskinesia, mainly triggered by exercise and stress, were investigated in a retrospective multi-center investigation aiming to characterize the clinical phenotype and its underlying molecular etiology. Three dogs were closely related and their pedigree suggested autosomal dominant inheritance. Laboratory diagnostic findings included mild lactic acidosis and lactaturia, mild intermittent serum creatine kinase (CK) elevation and hypoglycemia. Electrophysiological tests and magnetic resonance imaging of the brain were unremarkable. A muscle/nerve biopsy revealed a mild type II fiber predominant muscle atrophy. While treatment with phenobarbital, diazepam or levetiracetam did not alter the clinical course, treatment with a gluten-free, home-made fresh meat diet in three dogs or a tryptophan-rich, gluten-free, seafood-based diet, stress-reduction, and acetazolamide or zonisamide in the fourth dog correlated with a partial reduction in, or even a complete absence of, dystonic episodes. The genomes of two cases were sequenced and compared to 654 control genomes. The analysis revealed a case-specific missense variant, c.1658G>A or p.Arg553Gln, in the *PCK2* gene encoding the mitochondrial phosphoenolpyruvate carboxykinase 2. Sanger sequencing confirmed that all four cases carried the mutant allele in a heterozygous state. The mutant allele was not found in 117 Shetland Sheepdog controls and more than 500 additionally genotyped dogs from various other breeds. The p.Arg553Gln substitution affects a highly conserved residue in close proximity to the GTP-binding site of PCK2. Taken together, we describe a new form of paroxysmal exercise-induced dyskinesia (PED) in dogs. The genetic findings suggest that PCK2:p.Arg553Gln should be further investigated as putative candidate causal variant.

## 1. Introduction

Paroxysmal movement disorders are a group of diverse neurological conditions characterized by the episodic occurrence of involuntary movements. In most cases with such disorders, patients have a normal interictal examination [1]. In human medicine, paroxysmal movement disorders are classified into paroxysmal dyskinesias (PxDs) and episodic ataxias (EAs). The PxDs are further subdivided into four related forms, paroxysmal kinesigenic dyskinesia (PKD), paroxysmal non-kinesigenic dyskinesia (PNKD), paroxysmal hypnogenic dyskinesia (PHD), and paroxysmal exercise-induced dyskinesia (PED) [1].

PED in humans is most frequently due to genetic variants in the *SLC2A1* gene encoding the GLUT1 transporter mediating glucose transfer across the blood–brain barrier [1,2,3,4]. Dominant and recessive forms of *SLC2A1* related PED have been described. Depending on the specific variant, the PED may occur isolated [2,3] or in combination with other phenotypes, such as epilepsy, delayed development and mental retardation [4]. In other human patients with isolated or syndromic PED forms, genetic variants in *GCH1* or *PARKN* have been described [5,6].

In veterinary medicine, several breed-specific episodic movement disorders characterized by spasticity have been reported [7]. Their etiopathophysiology is heterogenous in different breeds and the causal genetic variants are only partially known. The so-called Scottie Cramp in Scottish Terriers was already recognized 50 years ago and is characterized by generalized or hind limb spasticity. The molecular cause has not yet been reported in the scientific literature [8,9,10]. Related phenotypes also with unclear causative genetic defects were reported in Bichon Frisé [11,12], Border Terriers [13,14,15], Boxers [16], Chinooks [17], German Shorthair Pointers [18], Jack Russell Terriers [19] and Maltese dogs [20].

In Soft-Coated Wheaten Terriers, an autosomal recessive paroxysmal dyskinesia is caused by a variant in the *PIGN* gene encoding the phosphatidylinositol glycan anchor biosynthesis class N (OMIA 002084-9615) [21]. Episodic falling syndrome in Cavalier King Charles Spaniels is an autosomal recessive disorder caused by variants in the *BCAN* gene encoding the brain-specific extracellular matrix proteoglycan brevican (OMIA 001592-9615) [22,23].

In this manuscript, we describe the clinical and diagnostic findings, treatment and outcome of four Shetland Sheepdogs with a paroxysmal movement disorder classified as PED together with our efforts to elucidate the underlying causative genetic defect. The study was conducted as a retrospective multi-center investigation.

## 2. Materials and Methods

### 2.1. Ethics Statement

All animal experiments were performed according to local regulations. All dogs in this study were privately owned and examined with the consent of their owners. The "Cantonal Committee for Animal Experiments" approved the collection of blood samples (Canton of Bern; permit 75/16).

### 2.2. Animal Selection

This study included four PED affected Shetland Sheepdogs, one from Germany and three related cases from the Netherlands (Figure 1). In the Dutch family, one of the cases is the mother of the other two cases, who are full siblings. A half sibling to the two cases with similar clinical signs died before blood samples could be drawn. For the genetic analyses, we used 117 additional blood samples of Shetland Sheepdogs without any reports of neurological disease and 515 dogs of various other breeds, which had been donated to the Vetsuisse Biobank. Additional details on the samples are given in Appendix A.

### 2.3. Clinical Examinations

Clinical examinations were performed at the University of Veterinary Medicine Hannover, Foundation, Utrecht University, Department of Clinical Sciences of Companion Animals, AniCura Veterinary Clinic Trier GbR, and Veterinary Clinic Neandertal GbR. All examinations were performed after written informed owner´s consent according to the ethical guidelines of the University of Veterinary Medicine Hannover, Foundation and Utrecht University. A resident or diplomate of the European College of Veterinary Neurology performed the examinations in every dog.

Stress tests were performed to aggravate clinical signs by playing with the dogs for up to 30 min or by applying different external stressful stimuli (*n* = 3). Heart rate, rectal body temperature, blood glucose, lactate, creatinine kinase (CK) and electrolytes were measured before and after playing.

### 2.4. Laboratory Examinations

Blood examinations were performed immediately after blood sampling and included blood cell count (ADVIA 120 Hematology System, Siemens Healthcare GmbH, Erlangen, Germany), biochemistry (Cobas c 311 analyzer, Roche Deutschland Holding GmbH, Mannheim, Germany) and electrolytes (RAPIDLab 1260, Siemens Healthcare GmbH). Plasma was examined for total thyroxine content via electro-chemiluminescence immunoassay (ECLI, *n* = 3). Serum for insulin measurements was frozen at −4 °C immediately after sampling and was examined within 24 h via chemiluminescent Immunoassay (CLIA, *n* = 1, Biocontrol, Ingelheim am Rhein, Germany). Urine samples were taken via cystocentesis and immediately frozen at −20 °C until examination. Analysis of urinary organic acids (*n* = 3) was performed by gas chromatography and mass spectroscopy (Biocontrol, and Biochemical Genetics Laboratory, San Diego, CA, USA). For detailed case information, see Appendix A.

### 2.5. Cardiac Examinations

Cardiac sonography was performed in an awake state in all 4 cases [24]. Ambulatory electrocardiography (ECG) was performed for 24 hours in case 4 with a bipolar triaxial lead system via telemetric ECG on a holter (Televet 100 Version 4.2, Engel Engineering Service GmbH, Heusenstamm, Germany) [25].

### 2.6. Muscle Examinations and Histopathology

Electrodiagnostic examinations of the axial and abaxial muscles and peripheral nerves in all 4 cases were performed using a Vicking Quest electrodiagnostic device (Nicolet Viking Quest IV, Nicolet EBE GmbH, Kleinostheim Germany). Recordings of the compound muscle action potentials (CMAP) and measurement of the motor nerve conduction velocity (mNCV) and the amplitude of the CMAP as well as repetitive nerve stimulation were performed with 0.5, 2, 3, 10 and 30 Hz of the radial and peroneal nerves. Muscle and nerve biopsies were taken from the extensor carpi radialis and tibialis cranialis muscles of case 4 under general anesthesia according to Platt and Olby [26] and were sent to the Clinical and Comparative Neuropathology Laboratory of the Ludwig-Maximilians-Universität, Munich. Samples underwent routine cryohistological processing, including enzyme histochemistry for cytochrome oxidase and nicotinamide adenine dinucleotide tetrazolium reductase, myofiber typing and special stains for the detection of polysaccharides (periodic acid Schiff) and lipids (oil red O), mitochondria and protein aggregates (Engel´s modified Gomori stain). Further samples of both muscles were subjected to transmission electron microscopy following glutaraldehyde fixation, embedding in epoxy resin, ultrasectioning and contrasting with lead citrate and uranyl acetate.

### 2.7. Additional Diagnostic Examinations

Electroencephalography (EEG, NicoletOne nEEG, Nicolet) of case 4 was obtained in an awake state with a montage according to Brauer et al. [27].

Low field magnetic resonance imaging (MRI) of the brain of case 4 was performed by the referring veterinarian. T2weighted (T2w), T1weighted (T1w) and fluid attenuation inversion recovery (FLAIR) sequences were available for review.

Cerebrospinal fluid (CSF) was sampled from case 4 in general anesthesia from the cisterna magna and was immediately examined for protein (Cobas c 311 analyzer) and cell content [26].

### 2.8. Autoantibodies

Serum and CSF of case 4 were screened for known and novel nervous system autoantibodies with cell-based assays (GAD65, NMDAR, GABABR, AMPARI, AMPAR2, DPPX, LGl1, CASPR2, GlyR, mGlu5) and immunofluorescence test (IFT) on mice hippocampi (Epilepsiezentrum Bethel, Bielefeld, Germany).

### 2.9. Fibroblast Culture

Dermal fibroblasts from case 4 were cultured to further examine mitochondrial function according to [28]. A skin biopsy as a starting material was taken from case 4 under local anesthesia (Lidocain 2 mL subcutaneously with 2 cm of spatial distance to the biopsy site; Lidocard 2% Mini-Plasco; B. Braun Melsungen AG, Melsungen, Germany).

### 2.10. Tryptophan Content of Therapeutic Diet

Tryptophan contents of a conventional gluten-containing diet (Markus Mühle^®^, Langenhahn, Germany), a conventional gluten-free diet (Wildkind^®^, Das Futterhaus-Franchise GmbH & Co. KG, Elmshorn, Germany) and a gluten- and grain-free, seafood-based diet (Purizon Fisch^®^, Matina GmbH, München, Germany) fed to case 4 were measured (Routine Laboratory, University of Rostock).

### 2.11. Whole Genome Sequencing of Two Affected Shetland Sheepdogs

Genomic DNA was isolated from the EDTA blood of affected dogs and healthy controls with the Maxwell RSC Whole Blood Kit using a Maxwell RSC instrument (Promega, Dübendorf, Switzerland). Illumina TruSeq PCR-free DNA libraries with 350 bp insert size of one affected Shetland Sheepdog from each of the two families were prepared (cases 2 and 4). We collected 331 and 321 million 2 × 150 bp paired-end reads on a NovaSeq 6000 instrument (37.4× and 38.0× coverage). Mapping and alignment were performed as described [29]. The sequence data were deposited under the study accession PRJEB16012 and the sample accessions SAMEA104091573 and SAMEA4867921 at the European Nucleotide Archive.

### 2.12. Variant Calling

Variant calling was performed as described [29]. To predict the functional effects of the called variants, the SnpEFF [30] software together with NCBI annotation release 105 for CanFam 3.1 was used. For variant filtering, we used 654 control genomes, which were publicly available. The control genomes were derived from 648 dogs of genetically diverse breeds and 8 wolves (Appendix A).

### 2.13. Gene Analysis

We used the dog CanFam 3.1 reference genome assembly for all analyses. Numbering within the canine *PCK2* gene corresponds to the NCBI RefSeq accessions XM_537379.6 (mRNA) and XP_537379.2 (protein).

### 2.14. Sanger Sequencing

To genotype the *PCK2*:c.1658G>A variant, a 468 bp PCR product was amplified from genomic DNA using the AmpliTaqGold360Mastermix (Thermo Fisher Scientific, Waltham, MA, USA) together with primers 5′-GCT ACA ACT TTG GGC GCT AC-3′ (Primer F) and 5′- ATG AGG GGT AGG AAG GGA TG-3′ (Primer R). After treatment with exonuclease I and alkaline phosphatase, amplicons were sequenced on an ABI 3730 DNA Analyzer (Thermo Fisher Scientific, Waltham, MA, USA). Sanger sequences were analyzed using the Sequencher 5.1 software (GeneCodes, Ann Arbor, MI, USA).

## 3. Results

### 3.1. Clinical Examinations and Family History

Four female Shetland Sheepdogs (age 2–6 years) were presented due to progressive dyskinetic episodes. Three of the four dogs were closely related, suggesting an inherited disorder. Although not conclusive, the pedigrees were compatible with an autosomal dominant mode of inheritance (Figure 1).

The episodes were characterized by generalized ataxia with hypermetria and muscular hypertonia of all limbs, dystonia, normal to mildly reduced mentation, and a mild tremor. In the more severe episodes, the dogs were no longer ambulatory. No signs of autonomic dysfunction were visible (normal size of pupils, no salivation, no defecation or urination, no signs of increased gastro-intestinal motility) (Video S1). The episodes varied from minutes to hours and could start while at rest, or during activity. In case 4, they were triggered by excitement or stress, like playing or after being startled by noise, according to the owner. Episodes in this dog were more intense and of longer duration after physical exercise. In cases 1–3, hot weather seemed to aggravate the clinical signs.

General and neurological examination was normal in all dogs, except for mild generalized muscle atrophy in case 4. Stress tests led to intermittent stiff gait in cases 1 and 2 but could not provoke a dystonic episode in case 4. The findings were consistent with a movement disorder or paroxysmal dyskinesia. Encephalopathy or neuromuscular disorders were also considered. All clinical examination results are summarized in Appendix A.

### 3.2. Laboratory Examinations

Interictal blood cell count, biochemistry, and electrolytes were mostly within the reference range in all four affected dogs. Mildly increased creatine kinase activity (CK) was seen in cases 2 and 3 (*n* = 2/4, 221 U/l and 350 U/l, respectively, reference: <220 U/l). Total thyroxine (tT4) was normal in all tested dogs (*n* = 3/3, Appendix A).

In case 4, fasted blood glucose was at the lower boundary of the reference range (72 mg/dL, reference: 70–110 mg/dL) and lactate at the upper reference limit (21.7 mg/dL, reference: 4.5–22.5 mg/dL). In venous blood, the base excess (BE) was −8.5 mmol/L (reference: −4 to 4 mmol/L) with a HCO_3_^−^ of 13 mmol/L (reference: 20–30 mmol/L), decreased pCO_2_ (19.7 mm Hg, reference: 35–55 mm Hg) with normal pH (7.44, reference: 7.3–7.45). These findings are consistent with a metabolic acidosis with respiratory compensation. Blood examination after the stress test showed increased CK values in cases 1 and 2 (*n* = 2/3), and mildly decreased blood glucose (69 mg/dL) and the deterioration of BE (−10 mmol/L) in case 4 (*n* = 1/3). The parallel insulin measurement in case 4 was normal (6.8 µU/mL, reference: 5–25 µU/mL; glucose-insulin-ratio 10.14, reference: <30).

Urinary organic acid analysis an showed increased excretion of lactate (*n* = 3/3) and 2-hydroxybutyrate (*n* = 1/3) or 3-hydroxybutyrate (*n* = 2/3) compared to normal control (Figure 2). A full overview of all clinical and diagnostic findings in the four affected dogs is given in Appendix A.

### 3.3. Additonal Examinations

Cardiac sonography and ECG were normal between episodes. ECG immediately before an episode revealed normal sinus rhythm with mild tachycardia (120–180 bpm). When a dyskinetic episode started, ECG was overlapped by muscle artefacts. ECG did not reveal cardiac pathology (Appendix A).

EEG in case 4 while the dog was awake showed mostly muscle artefacts and otherwise predominantly normal low voltage beta-rhythm.

Low-field MRI of the brain was performed in case 4 by the referring veterinarian and was available for review. No pathological abnormality was visible.

A CSF tap was performed in case 4 and was unremarkable: no cells were present, protein content was 11.8 mg/dL (reference <25 mg/dL), and glucose was 69 mg/dL (reference 42–77 mg/dL).

Electromyography (EMG) showed normal insertional discharge in all examined muscles without pathological spontaneous activity. Nerve-conduction studies showed normal nerve conduction velocities with normal CMAP amplitudes in all four dogs.

A search for autoantibodies in case 4 did not reveal any pathological findings.

### 3.4. Histopathology

Histopathological muscle changes in case 4 were sparse and featured a diffuse type II fiber predominant muscle atrophy. There was no evidence of mitochondrial changes and/or substrate accumulation on histology, enzyme histochemistry, or electron microscopy.

### 3.5. Clinical Management

Treatment with oral phenobarbital up to 3 mg/kg twice daily (cases 1–4) and levetiracetam 60 mg/kg three times daily (case 4) did not decrease the frequency of episodes according to the owners. Diazepam 2 mg/kg rectally did not seem to decrease the length of the episodes (case 4). Acetazolamid 50 mg/kg three times daily initially was reported to improve the clinical signs in case 4, but only for 6 months when the frequency started to increase again. Changing to zonisamide 10 mg/kg three times daily seemed to have an impact on episode frequency in the long term.

Supplementation with L-carnitine and multivitamins did not improve clinical signs in any of the four affected dogs.

A number of various commercial diets and gluten-free diets were tried but appeared to be unsuccessful in three dogs (case 1–3). However, in case 4, the feeding of a gluten-free, home-made fresh meat diet improved clinical signs. In case 4, the frequency of episodes improved with a commercially available gluten-free diet. Interruption of this diet led to the recurrence of an increased frequency of the episodes. A further change in diet to a seafood-based, gluten- and grain-free diet markedly improved the clinical signs again, even better than before. The last diet had the highest tryptophan content (Table 1).

Additional supplementation with tryptophan 50 mg/kg twice daily in case 4 combined with the prevention of stress and exhausting exercise further decreased clinical signs according to the owner. A summary of treatments in all four cases is given in Appendix A.

### 3.6. Outcome

We were able to monitor all four affected dogs over a period of 5 to 12 years and the dogs remained stable (Appendix A). Two dogs were event-free, one dog suffered from only one to two events per year. Exercise- and stress-management in combination with zonisamide, a seafood-based, gluten-free diet and tryptophan helped the fourth dog to stabilize on a low-level frequency of episodes. The dog continued to display one short episode every 2–3 days, during which she was still ambulatory (Figure 3). Six years after diagnosis, case 3 died of an acute renal failure, case 1 died of old age at 15 years of age. Cases 2 and 4 were still alive at the time of writing (Appendix A).

### 3.7. Genetic Analysis

We sequenced the genomes of cases 2 and 4 at 37.4× and 38.0× coverage and called SNVs and short indels with respect to the CanFam 3.1 reference genome assembly. We then searched for shared heterozygous and homozygous variants in the genome sequence of the two affected dogs that were not present in 654 control genomes. This analysis yielded 1066 variants that were exclusively shared by the two cases and not present in any of the control genomes (Appendix A). None of the shared homozygous variants were predicted to have a protein-changing effect, but 10 heterozygous variants shared by the two cases were predicted to be protein changing (Table 2).

We then prioritized the 10 private protein-changing variants according to the functional knowledge on the altered genes. We considered the *PCK2* gene encoding the mitochondrial phosphoenolpyruvate carboxykinase 2 the most likely candidate gene for the observed clinical phenotype (Table 3).

The top candidate variant on the genomic level was Chr8:4,107,413G>A. The corresponding variant designations on the cDNA and protein level are XM_537379.6:c.1658G>A or XP_537379.2:p.(Arg553Gln), respectively. We confirmed the variant by Sanger sequencing. (Figure 4A). The variant is predicted to alter a positively charged arginine close to the GTP-binding site of PCK2 into a neutral glutamine without major changes to the threedimensional structure of the enzyme. The wildtype arginine is strictly conserved across animals from *C. elegans* up to mammals (Figure 4B).

We then genotyped all four affected dogs, 117 Shetland Sheepdog controls without any movement disorder or seizures and 515 control dogs from 71 genetically diverse dog breeds for the PCK2:c.1658G>A variant and found a perfect genotype-phenotype association. All four cases carried one copy of the mutant allele, while all control dogs were homozygous for the reference allele (Table 4; Appendix A). The segregation of the genotypes in the available family was compatible with an autosomal dominant inheritance (Figure 1). An attempt to establish a fibroblast culture from case 4 to perform analyses on mitochondrial function was unsuccessful. While fibroblasts from a healthy control dog grew as expected, fibroblasts from case 4 did not grow appropriately to allow biochemical studies.

## 4. Discussion

The present article describes the clinical and diagnostic findings of an inherited PED in four Shetland Sheepdogs with a suspected deficiency in the mitochondrial phosphoenolpyruvate carboxykinase 2 (PCK2). There are two isoforms of phosphoenolpyruvate carboxykinase, a cytosolic isoform, encoded by the *PCK1* gene, and a mitochondrial isoform, encoded by the *PCK2* gene. The tissue specificity of these isoforms has been described [32]. PCK1 is hormonally regulated, with insulin switching the enzyme off, whereas PCK2 does not seem to be regulated by hormones, but rather by mitochondrial GTP levels [32,33,34,35].

Our genetic analysis revealed a *PCK2* missense variant in the affected dogs, which was predicted to change an evolutionarily conserved amino acid located close to the GTP-binding site of PCK2 [31]. The mutant allele was exclusively found in a heterozygous state in the four studied PED cases and absent in more than one thousand control dogs. As PCK2 is a monomeric enzyme, we consider it unlikely that the p.Arg553Gln substitution will have a dominant negative effect. We speculate that haploinsufficiency during periods of high energy demand may be causing the phenotype in the affected dogs. We have to caution that our genetic analysis was strictly based on the assumption of a shared causative genetic variant between the two sequenced cases. If the phenotypes in the studied dogs are due to different genetic and/or environmental causes, the detected PCK2 variant might be functionally neutral. We also have to caution that our bioinformatics analysis considered only small genetic variants and would not have detected any large structural variants involving more than ~25 consecutive nucleotides.

We did not succeed in proving the functional relevance of the genetic variant by enzyme activity assay as fibroblasts from a PED-affected dog did not proliferate appropriately, whereas a parallel culture of control fibroblasts from a healthy dog grew sufficiently. We have observed this phenomenon before in some human diseased fibroblasts (personal observation, A.M.D.).

Few human patients with primary phosphoenolpyruvate carboxykinase deficiency have been reported in the literature, mainly with isolated variants in PCK1 [36,37,38]. These patients suffer from liver dysfunction, sometimes leading to liver failure, hypoglycemia, lactic acidosis and sometimes complex symptoms [36,37,38]. To the best of our knowledge, no PCK2 variants have been described in human patients. Some reports in the older literature claim a deficiency of mitochondrial phosphoenolpyruvate carboxykinase in human patients [39,40]; however, later on, these diagnoses were withdrawn or revised. One publication reports a patient with a complex phenotype suffering from both cytosolic and mitochondrial phosphoenolpyruvate carboxykinase deficiency [41].

The current human genome and exome data of the gnomAD browser do not indicate any intolerance of heterozygous *PCK2* loss of function variants [42,43]. Interestingly, the variant found in the affected Shetland Sheepdogs, p.Arg553Gln, also represents a rare variant in humans. The gnomAD data lists 27 heterozygotes for the ^553^Gln-allele, which has a frequency of 9.55 × 10^−5^ in the dataset [43]. It is unknown whether these persons have any clinical phenotype.

In the four affected dogs, paroxysmal hypertonic dyskinetic episodes often being triggered by stress, excitement or hot weather were the prominent clinical signs. Transient hypoglycemia, lactaturia, ketonuria and subsequent metabolic acidosis were noted. A muscle biopsy showed mild type II fiber predominant muscle atrophy, which is common in metabolic disease [44]. No cardiac abnormalities, structural changes of brain parenchyma, or signs of hepatopathy were seen in our canine patients. Similar clinical signs were described in a Shetland Sheepdog in 1992, but at this time the etiopathology remained obscure [45].

We suggest that decreased PCK2 activity may have led to impaired gluconeogenesis and energy metabolism in the affected Shetland Sheepdogs. The most severely affected dog (case 4) showed borderline low glucose. We suggest that clinical signs are therefore most pronounced in times of stress or exercise, when the body has an increased demand for energy. The resulting shortage of energy might first affect those organs with a high energetic turnover, such as muscles and the brain. This might result in the paroxysmal dyskinetic events seen in the Shetland Sheepdogs similar to glucose transporter GLUT1 deficiency where gait abnormalities are observed [46]. Phosphoenolpyruvate carboxykinase is expressed in astrocytes [47] and impaired gluconeogenesis may have a similar effect. GTP levels in neuronal cells may be altered, which may affect the production of tetrahydrobiopterin [48] and thus synthesis of neurotransmitters and NO.

As not all dogs had measurable episodes of hypoglycemia, another albeit highly speculative pathomechanism should be considered: PCK2 acts as a sensor of the citric acid cycle (‘Krebs cycle’) flux by removing oxaloacetate [34]. A reduced PCK2 activity could result in the accumulation of oxaloacetate hampering the citric acid cycle flux. PCK2 is the only isoform providing phosphoenolpyruvate carboxykinase activity in pancreas and possibly other tissues, linking the production of mitochondrial GTP to anaplerotic phosphoenolpyruvate cycling [35]. PCK2 also has pyruvate kinase activity, which theoretically could enhance mitochondrial pyruvate formation and transformation into oxaloacetate when energy levels in the mitochondrion are low, thus contributing to the citric acid cycle, a situation suggested to be stress-related [49]. We have to caution that our two mechanistic hypotheses so far are not supported by any experimental data. As heterozygous PCK2 loss-of-function variants have not yet been identified as causative for a corresponding phenotype in human patients, functional validation will be of the utmost importance to further evaluate the hypothetical link of canine PCK2 deficiency and PED.

In human medicine, a ketogenic diet with a low glycemic index is recommended for diseases that affect energy metabolism to avoid high insulin peaks [50,51]. In humans, ketogenic diets classically include high amounts of fat and low amounts of carbohydrates, which forces the body to change to a ketogenic metabolism, to provide ketone bodies as an alternative energy source for the brain and muscles [52,53]. In contrast, dogs do not tend to change into a ketogenic metabolism as easily as humans [54]. Additionally, a diet with a high amount of fat may cause pancreatitis in dogs [55]. Therefore, it is common practice in veterinary medicine to use a gluten- and grain-free diet as a more compatible form of a diet with a low glycemic index. This was also successfully applied in the presented Shetland Sheepdogs, whose clinical signs improved after being fed a gluten-free, high-protein diet. In one dog, a seafood-based diet showed the best results. This diet was rich in tryptophan, which is an essential amino acid involved in the production of serotonin [56]. Increased serotonin levels in the brain reduce stress [57]—an important trigger of dyskinetic episodes observed in this dog. Tryptophan may enhance de novo synthesis of NAD [58], an important metabolic regulator for energy metabolism. However, these mechanisms are highly speculative as we did not measure tryptophan nor neurotransmitter levels. Additionally, in one dog, antiepileptic drugs of the class of carbonic anhydrase inhibitors (acetazolamide or zonisamide) were used and correlated with improved clinical signs. Successful treatment with acetazolamide has been described in Soft-coated Wheaten Terriers and Golden Retrievers with paroxysmal dyskinesia [59,60]. The exact mechanism of action is not clear yet, but it is thought that acetazolamide supports ion transport across the blood–brain barrier, modifying the intracellular pH and, therefore, the transmembrane potential, which lowers the excitability of neurons [51].

## 5. Conclusions

We describe a new PED with presumed autosomal dominant inheritance in Shetland Sheepdogs. Stress management, a specific diet and pharmacological therapy resulted in the partial or complete suppression of hypertonic dyskinetic episodes and enabled a good quality of life. The genetic analysis suggested that the *PCK2*:p.Arg553Gln missense variant should be considered and further evaluated as potential candidate causal variant for this phenotype. This study provides an interesting potential link between exercise-induced hypoglycemia, mitochondrial energy metabolism and paroxysmal dyskinesia that warrants further investigation.

## Figures and Tables

**Figure 1 genes-11-00774-f001:**
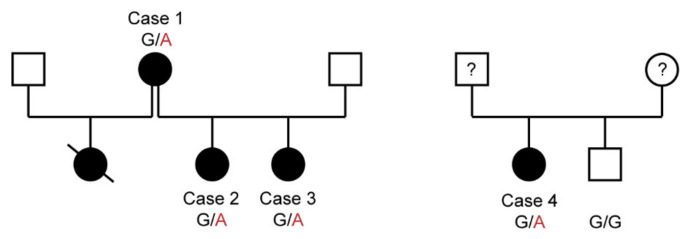
Pedigrees of the four affected Shetland Sheepdogs. Squares represent males and circles represent females. Filled symbols indicate affected dogs and open symbols indicate non-affected dogs. Symbols with question marks indicate dogs of an unknown phenotype. The strike-through symbol represents a dog that died before the beginning of the investigation. According to the owner, this dog had similar clinical signs as the other affected Shetland Sheepdogs. The genotypes at the *PCK2*:c.1658G>A variant are given for dogs, from which samples were available (see Section 3.5).

**Figure 2 genes-11-00774-f002:**
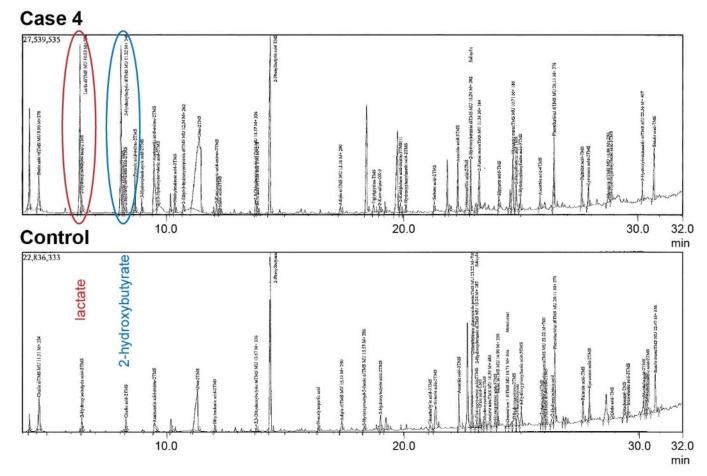
Metabolic screening of urinary organic acids in case 4. Gas chromatographic examination of a 3-year-old paroxysmal exercise-induced dyskinesia (PED)-affected dog’s urine showed an increased excretion of lactate (red circle) and 2-hydroxybutyrate (blue circle) compared to a healthy control dog.

**Figure 3 genes-11-00774-f003:**
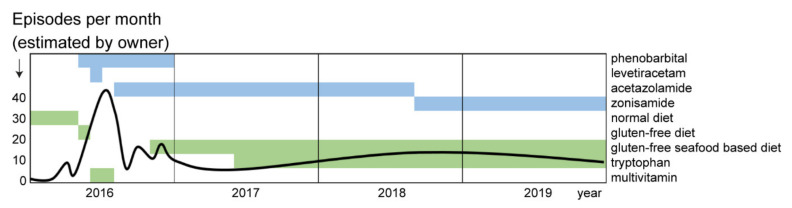
Frequency of dyskinetic episodes in case 4. The estimated frequency of episodes per month (black line), contemporaneous medications and diets are displayed with green and blue bars, respectively. A combination of seafood-based, gluten-free diet with supplementation of tryptophan and treatment with acetazolamide or zonisamide seemed to reduce frequency of episodes.

**Figure 4 genes-11-00774-f004:**
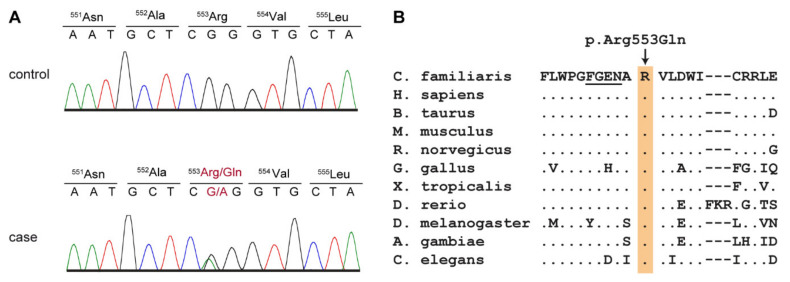
Details of the PCK2:c.1658G>A variant. (**A**) Representative Sanger sequencing electropherograms of two dogs with the different genotypes are shown. (**B**) Evolutionary conservation of the arginine residue at position 553 of the PCK2 protein. A multiple species alignment illustrates that this residue is strictly conserved across animals from worms to mammals. Amino acids 548-551 form part of the GTP-binding site of PCK2 and are underlined [31].

**Table 1 genes-11-00774-t001:** Tryptophan content in three different commercially available food samples and owners’ subjective perception of clinical signs.

Diet	Dry Matter	Tryptophan	Clinical Response
normal	921 g/kg	1.52 g/kg	worsening of clinical signs
gluten-free	950 g/kg	1.65 g/kg	improved clinical signs
seafood gluten- and grain free	951 g/kg	2.97 g/kg	markedly improved clinical signs

**Table 2 genes-11-00774-t002:** Shared variants in the two sequenced cases.

Filtering Step	Heterozygous Variants	Homozygous Variants
Case-specific variants	1030	36
Case-specific protein-changing variants	10	0

**Table 3 genes-11-00774-t003:** Heterozygous protein-changing variants shared by the 2 PED cases and absent in 654 controls.

Gene	Protein	Variant
*C7*	complement C7	p.Glu799*
*CDH24*	cadherin 24	p.Asp669Asn
*CNNM1*	cyclin and CBS domain divalent metal cation transport mediator 1	p.Asp170Tyr
*LOC485317*		p.Lys270Asn
*LOC106559343*		p.Ala179Gly
*LOC111097338*		p.Arg182Trp
*OR2A14*	olfactory receptor family 2 subfamily A member 14	p.Ser56Arg
*OR2A14*	olfactory receptor family 2 subfamily A member 14	p.Leu54Gln
*PCK2*	phosphoenolpyruvate carboxykinase 2, mitochondrial	p.Arg553Gln
*TM9SF1*	transmembrane 9 superfamily member 1	p.His362Tyr

**Table 4 genes-11-00774-t004:** Association of the genotypes at *PCK2*:c.1658G>A with paroxysmal dyskinesia.

Dogs	G/G	G/A
Cases Shetland Sheepdogs (*n* = 4)	0	4
Controls Shetland Sheepdogs (*n* = 117)	117	0
Controls other breeds (*n* = 515) ^1^	515	0

^1^ Independent from the 654 controls of the variant discovery.

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
