# Peer review of "Mitochondrial PCK2 Missense Variant in Shetland Sheepdogs with Paroxysmal Exercise-Induced Dyskinesia (PED)"

_genes, 2020, doi:10.3390/genes11070774_

Round 1

Reviewer 1 Report

The study from Nessler et al. investigated a pedigree including four Shetland Sheepdogs affected with Paroxysmal Exercise-Induced Dyskinesia. They conducted laboratory diagnostic, clinical and genetic assessments.

The manuscript is well written, detailed, and the results are relevant to the field. I do not see any methodological issue. The genomic workflow is not fully reported in the current manuscript, but it is well described in a previous study. I have just few minor comments:

  1. At the beginning of the abstract and at the end the introduction the authors should briefly outline the study design. For example, the abstract is very detailed and well written, but since the second sentence they started describing the results. Few introductory sentences about the study design would help the reader.
  2. In the abstract, the authors wrote all four cases carried the mutant allele. They stated two cases were sequenced using next generation sequencing, but they should specific the candidate variant was assessed in the other two cases by Sanger sequencing.
  3. Please specify in the abstract the 654 control genomes were from different breeds.
  4. In the methods it is not easy to understand how many samples were characterized for the clinical and diagnostic assessment (I assume all four cases). For instance, in section 2.2 please briefly specify for what all the blood samples were used (clinical, genomics, etc). Please mention also how many urine samples were collected. The Additional Diagnostic Examination (2.7) including the CSF collection was conducted only on case 4?. In how many cases the Tryptophan content was measured? I would suggest to add some reminders also in the results, it would be easier for the reader.
  5. How was conducted the selection of cases 2 and 4 for whole genome sequencing?

Author Response

(1)

At the beginning of the abstract and at the end the introduction the authors should briefly outline the study design. For example, the abstract is very detailed and well written, but since the second sentence they started describing the results. Few introductory sentences about the study design would help the reader.

Response: We added the requested information to the abstract and the end of the introduction.

(2)

In the abstract, the authors wrote all four cases carried the mutant allele. They stated two cases were sequenced using next generation sequencing, but they should specific the candidate variant was assessed in the other two cases by Sanger sequencing.

Response: Revised accordingly.

(3)

Please specify in the abstract the 654 control genomes were from different breeds.

Response: We agree with the reviewer that this information is relevant. However, in order not exceed the 250 word limit for abstract text that is included in Pubmed records, we decided not to include this information in the abstract. We added the information to the methods and the breeds are listed in detail in Table S2.

(4)

In the methods it is not easy to understand how many samples were characterized for the clinical and diagnostic assessment (I assume all four cases). For instance, in section 2.2 please briefly specify for what all the blood samples were used (clinical, genomics, etc). Please mention also how many urine samples were collected. The Additional Diagnostic Examination (2.7) including the CSF collection was conducted only on case 4?. In how many cases the Tryptophan content was measured? I would suggest to add some reminders also in the results, it would be easier for the reader.

Response: We understand the confusion to the reader which is caused by not entirely consistent examinations due to the retrospective and multi-centric nature of the diagnostic workup. This is the main reason for the inclusion of Table S3, which transparently documents what has been done in each case. We agree, that additional information within the text might be helpful for a better understanding. We added the number of performed tests in material and methods and in the results part in multiple instances.

(5)

How was conducted the selection of cases 2 and 4 for whole genome sequencing?

Response: We chose two cases for sequencing in order to have one affected dog from each of the two families. We added this rationale to the methods section.

Reviewer 2 Report

This small case series is well written and describes the spectrum of clinical testing used to understand a clinical disorder. The authors' are appropriately transparent about the limited functional consequences that can be inferred from the PCK2 missense variant alone. The primary concern this reviewer has regarding the manuscript is the portion of the discussion that postulates a mechanism/function without supporting evidence in the results (lines 353-380). While this information is very interesting and the authors' acknowledge that the mechanistic hypotheses are not supported by the results, please consider limiting this part of the discussion and reframe your hypotheses in the context of the study results.

Minor comments:

  1. Did authors visually (e.g. IGV) confirm all variants listed in Table 3? While sanger sequencing was done on the variant of interest, it was unclear how the variants in Table 3 were validated.
  2. In table three, please list the variant effect (e.g. missense mutation, etc).
  3. Please consider removing the methods description on fibroblast culture. The authors' state that this test did not work, and no results are presented. Methods should describe the results presented. The discussion paragraph eluding to the attempts at fibroblast culture seems the most appropriate and least confusing way to relay this information to the reader.

Author Response

(1)

This small case series is well written and describes the spectrum of clinical testing used to understand a clinical disorder. The authors' are appropriately transparent about the limited functional consequences that can be inferred from the PCK2 missense variant alone. The primary concern this reviewer has regarding the manuscript is the portion of the discussion that postulates a mechanism/function without supporting evidence in the results (lines 353-380). While this information is very interesting and the authors' acknowledge that the mechanistic hypotheses are not supported by the results, please consider limiting this part of the discussion and reframe your hypotheses in the context of the study results.

Response: As the reviewer correctly points out, we were not able to obtain functional data to prove our mechanistic hypotheses. We nonetheless think that it is important to formulate these hypotheses as they should be testable in future investigations. We thoroughly revised and shortened this section to keep the manuscript reasonably concise.

Minor comments:

(2)

Did authors visually (e.g. IGV) confirm all variants listed in Table 3? While sanger sequencing was done on the variant of interest, it was unclear how the variants in Table 3 were validated.

Response: We indeed confirmed all 10 protein-changing variants by visual inspection in IGV. As we had high coverage data (>30x) on both dogs, all the called variants looked very convincing in IGV. Only the PCK2 variant was confirmed by Sanger sequencing. The manuscript was not changed with respect to this comment.

(3)

In table three, please list the variant effect (e.g. missense mutation, etc).

Response: Do to journal style, it is not possible to add another column to table 3, otherwise this table would become too wide. The requested information is actually contained in the third column “variant”. Here, it can be clearly seen that the first variant is a nonsense variant, while the other nine variants represent missense variants.

(4)

Please consider removing the methods description on fibroblast culture. The authors' state that this test did not work, and no results are presented. Methods should describe the results presented. The discussion paragraph eluding to the attempts at fibroblast culture seems the most appropriate and least confusing way to relay this information to the reader.

Response: We removed the methods section on fibroblast culture as requested.

Reviewer 3 Report

The manuscript, Mitochondrial PCK2 Missense Variant in Shetland Sheepdogs with Paroxysmal Exercise-Induced Dyskinesia (PED) by Nessler J et al., describes a rare form of hypertonic dyskinesia in 4 Shetland sheepdogs and a WGS effort to determine a variant underlying the phenotype. It summarizes clinical and laboratory studies conducted over at least 12 years in the Netherlands and Germany. It is a well written exposition in nearly flawless English and appropriate medical and genetic terminology.

Given, however, the uncertainties of the findings (dominant transmission is not proven, unknown function of some genes in table 3, affect of the PEPCK2 variant on metabolism, the same mutation tolerated in humans), alternative hypotheses should be discussed, rather than the long unsupported metabolomic speculations in line 353-380. As said, further investigation is warranted.

This reviewer also recommends a few revisions to improve syntax and spelling.

Line 49, keyword mitochondrium should be 'mitochondrion'.

Line 54, The sentence is improved by placing a period after the word movements. This followed by, "In most such disorders, patients have a ...."

Line 142, (CSF) was sampled under (rather than in) general ...., and delete the awkward word "suboccipitally" which adds no more information than the cisterna magna location.

Figure 1, The relationships would be clearer if there were 2 lines descending from the symbol for case 1, one going to each horizontal line indicating a litter. The figure would also be more compact if the offspring symbols were arranged on the horizontal lines. Please delete the last sentence of the figure legend, which is an interpretive speculation, rather than a description of the figure.

Line 237, Should not the word Suboccipital be spelled Subocciputal? Is this even a word? It would be clearer to say simply, 'A CSF tap was performed....'; the methods section already indicates the tap location. 

Line 238, The reference level of CSF glucose as a percent of blood glucose is confusing since case 4 seems to have a level considerable above the reference level. Is there no comment about that finding?

Line 253, the word deteriorate suggests a decline or a decrease. Perhaps change that word to 'increase'.

Line 259, Because there are no statistics offered, please delete the word 'significantly', just say improved.

Figure 3 legend, delete the word 'potentially' from the last line.

Line 325-328, It would be much more satisfying to the reader if the authors would be more specific about how the putative mutation might alter the enzyme structure. It appears to be close to the GTP binding site in primary sequence, but wha can be predicted about the native conformation? Since it is quite rare for half-normal enzyme activity in a heterozygous individual to create a phenotype, is there a potential mechanism for a dominant negative effect?

Line 333, Delete the word 'Only'.

Line 342, 'orthologous' would be a better term to use than 'homologous' since the variants are identical and in the same genes.

Line 350, cardiological is awkward, use 'cardiac' instead.

References, The instructions to authors allow either listing all authors or listing the first 10 followed by et al., but one should be consistent. For instance, reference 2 lists 12 authors followed by et al, but reference 3 lists 16 authors. Reference 38, lists 25 authors. Pick a convention and use it throughout.

Author Response

(1)

Given, however, the uncertainties of the findings (dominant transmission is not proven, unknown function of some genes in table 3, affect of the PEPCK2 variant on metabolism, the same mutation tolerated in humans), alternative hypotheses should be discussed, rather than the long unsupported metabolomic speculations in line 353-380. As said, further investigation is warranted.

Response: We added several alternative hypotheses that do not require a causal role of the identified PCK2 variant to the second paragraph of the discussion. At the same time, we revised and shortened our metabolic hypotheses (see also comment 1 from reviewer 2).

(2)

Line 49, keyword mitochondrium should be 'mitochondrion'.

Response: Revised accordingly.

(3)

Line 54, The sentence is improved by placing a period after the word movements. This followed by, "In most such disorders, patients have a ...."

Response: Revised accordingly.

(4)

Line 142, (CSF) was sampled under (rather than in) general ...., and delete the awkward word "suboccipitally" which adds no more information than the cisterna magna location.

Response: Revised accordingly.

(5)

Figure 1, The relationships would be clearer if there were 2 lines descending from the symbol for case 1, one going to each horizontal line indicating a litter. The figure would also be more compact if the offspring symbols were arranged on the horizontal lines. Please delete the last sentence of the figure legend, which is an interpretive speculation, rather than a description of the figure.

Response: We revised the figure and the legend as requested.

(6)

Line 237, Should not the word Suboccipital be spelled Subocciputal? Is this even a word? It would be clearer to say simply, 'A CSF tap was performed....'; the methods section already indicates the tap location.

Response: Revised accordingly.

(7)

Line 238, The reference level of CSF glucose as a percent of blood glucose is confusing since case 4 seems to have a level considerable above the reference level. Is there no comment about that finding?

Response: We changed the definition of the reference range to “42-77 mg/dl”, which should be much easier to understand.

(8)

Line 253, the word deteriorate suggests a decline or a decrease. Perhaps change that word to 'increase'.

Response: Revised accordingly.

(9)

Line 259, Because there are no statistics offered, please delete the word 'significantly', just say improved.

Response: Revised accordingly.

(10)

Figure 3 legend, delete the word 'potentially' from the last line.

Response: Revised accordingly.

(11)

Line 325-328, It would be much more satisfying to the reader if the authors would be more specific about how the putative mutation might alter the enzyme structure. It appears to be close to the GTP binding site in primary sequence, but what can be predicted about the native conformation? Since it is quite rare for half-normal enzyme activity in a heterozygous individual to create a phenotype, is there a potential mechanism for a dominant negative effect?

Response: We thank the reviewer for this valuable comment. We performed structure modelling in Swiss-Model based on the template structure 3dt7.3.A. According to this analysis, the p.Arg553Gln variant will not induce any major conformational changes in the mutant protein. The side chain of the mutant Gln is only marginally smaller than the side chain of the wildtype Arg. However, the amino acid substitution will remove one positive charge and we speculate that the altered charge might cause the functional effect.

We added a brief statement about the three-dimensional structure to the results section. We further expanded the discussion regarding potential explanations for the observed dominant inheritance. Although uncommon, we still think that haploinsufficiency is more likely than a dominant-negative effect and at the same time provides a plausible explanation for the observed variability in phenotype severity between the cases.

(12)

Line 333, Delete the word 'Only'.

Response: Revised accordingly.

(13)

Line 342, 'orthologous' would be a better term to use than 'homologous' since the variants are identical and in the same genes.

Response: We think that “orthologous” is not appropriate as orthology implies that two genes have evolved from the same ancestral gene by speciation. This term can hardly be applied to variants. In the context of the manuscript, the p.Arg553Gln variants in humans and dogs are due to independent mutation events and the mutant alleles are not identical by descent.

To address the comment, we now deleted the phrase “homologous to”, which is also clearly not appropriate. The sentence now reads: Interestingly, the variant found in the affected Shetland Sheepdogs, p.Arg553Gln, also represents a rare variant in humans.

(14)

Line 350, cardiological is awkward, use 'cardiac' instead.

Response: Revised accordingly.

(15)

References, The instructions to authors allow either listing all authors or listing the first 10 followed by et al., but one should be consistent. For instance, reference 2 lists 12 authors followed by et al, but reference 3 lists 16 authors. Reference 38, lists 25 authors. Pick a convention and use it throughout.

Response: We revised reference 2 and hope that all references now contain complete listings of all authors. We ask the production department of MDPI to double-check this for consistency with the journal policy.